# Heme Oxygenase-1 Signaling and Redox Homeostasis in Physiopathological Conditions

**DOI:** 10.3390/biom11040589

**Published:** 2021-04-16

**Authors:** Valeria Consoli, Valeria Sorrenti, Salvo Grosso, Luca Vanella

**Affiliations:** Department of Drug and Health Sciences, Biochemistry Section, University of Catania, 95124 Catania, Italy; valeria_consoli@yahoo.it (V.C.); salvogrosso@outlook.it (S.G.); lvanella@unict.it (L.V.)

**Keywords:** heme oxygenase, Nrf2, GSH, TIGAR

## Abstract

Heme-oxygenase is the enzyme responsible for degradation of endogenous iron protoporphyirin heme; it catalyzes the reaction’s rate-limiting step, resulting in the release of carbon monoxide (CO), ferrous ions, and biliverdin (BV), which is successively reduced in bilirubin (BR) by biliverdin reductase. Several studies have drawn attention to the controversial role of HO-1, the enzyme inducible isoform, pointing out its implications in cancer and other diseases development, but also underlining the importance of its antioxidant activity. The contribution of HO-1 in redox homeostasis leads to a relevant decrease in cells oxidative damage, which can be reconducted to its cytoprotective effects explicated alongside other endogenous mechanisms involving genes like *TIGAR* (TP53-induced glycolysis and apoptosis regulator), but also to the therapeutic functions of heme main transformation products, especially carbon monoxide (CO), which has been shown to be effective on GSH levels implementation sustaining body’s antioxidant response to oxidative stress. The aim of this review was to collect most of the knowledge on HO-1 from literature, analyzing different perspectives to try and put forward a hypothesis on revealing yet unknown HO-1-involved pathways that could be useful to promote development of new therapeutical strategies, and lay the foundation for further investigation to fully understand this important antioxidant system.

## 1. Heme Oxygenase System Introduction

Heme oxygenase (HO) is the enzyme responsible for degradation of endogenous iron protoporphyirin Heme; it catalyzes the reaction’s rate-limiting step, producing ferrous ions, carbon monoxide (CO), and biliverdin (BV), whose reduction is operated by biliverdin reductase, resulting in bilirubin (BR) formation [1]. HO is also considered the main source of endogenous CO production.

The heme oxygenases are the only useful means for cell to catabolize heme, hence they are labeled as housekeeping genes and it is designated to control the cells heme pool [2].

In mammals, the microsomal heme oxygenase system consists of three enzymes: HO, NADPH cytochrome c (P450) reductase, and biliverdin reductase, and they employ stoichiometric quantities of NADPH and molecular oxygen for BR formation [3].

Heme oxygenase exists in three active isoforms, HO-1, HO-2, and HO-3. HO-1, also known as Heat shock protein 32 (Hsp32), is the inducible isoform whose expression can be upregulated by different stress conditions and activated by multiple stimuli, including UV irradiation, infections, heavy metals, nitric oxide, cytokines, and oxidized Low-Density Lipoprotein (LDL) [4]; HO-2 and HO-3 represent the constitutive isoforms expressed at basal levels in most of human tissues, in particular there are higher concentration levels of this isoforms in neurons, the spleen, and the liver [5].

HO-1 localization is in the endoplasmic reticulum (ER), anchored by a single transmembrane domain at the extreme carbonyl terminus, otherwise HO-2 is situated in the cellular microsomal fraction [6].

It has been reported that the HO-1 protein is also present in liver mitochondria, other than microsomes, where it regulates redox homeostasis depending on heme availability [7]. Recent studies have investigated the HO-1 subcellular localization proving its presence in other cell’s compartments such as smooth endoplasmic reticulum, nucleus, and mitochondria. It has been noticed that HO-1 migrates from cytosol into the nucleus where it mediates activation of oxidant-responsive transcription factor, which is independent from its enzymatic activity. The nuclear translocation was observed particularly under cellular stress conditions [8]. However, HO-1 nuclear activity is not completely understood yet.

HOs play a pivotal role in regulating redox homeostasis in virtue of their anti-inflammatory, antioxidant, and anti-apoptotic properties. In addition to its enzymatic activity, Vanella et al., described HO-1 non-canonical functions including chaperone activity mediated by protein-protein interactions, transcriptional regulation, intracellular compartmentalization, and immunomodulation [9]. HO-1 may also function as a pleiotropic regulator of inflammatory signaling pathways, exploiting its end-products activities. Heme degradation products are all biologically active molecules mostly implicated in tissue redox homeostasis and are generated under a multitude of pathological conditions [10,11,12].

## 2. HO-1 Transcriptional Regulation

Studies on *HMOX1* promoter reveals the existence of enhancer sequences responsible for the binding of redox-sensitive transcription factors, such as activator protein-1 (AP-1), nuclear factor-kappa B (NF-kB), Hypoxia inducible factor (HIF), and nuclear factor E2-related factor-2 (Nrf2) [13]. AP-1 is a dimer of Jun and Fos family proteins and it is considered a redox-sensitive transcription factor that senses and transduces changes in cellular redox status and modulates gene expression responses to oxidative and electrophilic stresses. The induction of HO-1 expression requires *AP-1* activation to respond to some oxidative inducers, such as lipopolysaccharide [14].

Modification of the cellular redox state leads to a change in the thiol status, which plays a key role in regulating NF-κB activation. In the basal state, NF-ĸB is sequestered as inactive form by its inhibitory protein IĸB that retains the heterodimer, composed of p50 and p65 subunits, in the cytoplasm. In the presence of specific stimuli, such as pro-inflammatory cytokines, oxidative stress, and growth factors, the IĸB kinase gets activated and phosphorylates IĸB, ultimately leading to its ubiquitination and degradation. As a consequence, NF-κB subunits p50 and p65 are free to translocate into the nucleus and regulate transcription of multiple target genes including HO-1 [15].

Hypoxic conditions induce HO-1 expression in cell and tissues and implicate HIF-1 in this response [16]. HIF-1 is a hetero-dimeric protein consisting of inducible α and constitutive β subunits, that contain N-terminal basic-helix-loop-helix-PAS domains required for dimerization and DNA binding. Activation of the MAPK pathway appears to specifically phosphorylates HIF-1α, which leads to its translocation to the nucleus and association to HIF-1α and p300/CBP. This complex binds to the HO-1 HRE site and induces its transcription. Nrf2 is the major *HMOX1* gene transcriptional regulator and it is capable of recognizing and binding the StRE/ARE sites [17]. Nrf2 is anchored in the cytoplasm by the Kelch-like ECH-associated protein (Keap1), which also functions as transcriptional inhibitor. Under basal conditions, Keap1 forms a complex with Nrf2 and prevents its nuclear translocation. Several lines of evidence allege oxygen reactive species implication in Nrf2- Keap1 bond disruption resulting in an increased Nrf2 accumulation in the nucleus, which led to speculation on Keap1 being a redox sensor [18,19], confirmed by further studies on Keap1 structure and Nrf2-Keap1 complex dynamics [20,21,22].

Cellular exposure to inducting stimuli leads to Nrf2 dissociation from Keap1 and subsequent translocation to the nucleus, where its transcription factor activity involves the formation of stable heterodimers with members of small Maf proteins family as MafF, MafG, and MafK [23].

The heme binding protein BTB and CNC homologue 1 (Bach1) has been identified as a negative transcriptional regulator of *HMOX1* gene. Bach1 forms a complex with small Maf proteins and competes with Nrf2 for StRE binding sites. Heme functions as inhibitor of Nrf2-DNA-binding activity through the formation of a complex with Bach1 and promotes its nuclear exportation [24,25]. Small Maf proteins act as dual-function transcription factors exploiting the exchange of their heterodimerization partners. Forming heterodimers with hematopoietic cell-specific p45 NF-E2 or NF-E2-related factors (Nrf), they activate antioxidative stress genes, e.g., HO-1, and in the meantime they repress the same genes through Bach1 interaction (Figure 1). Heme ability of inducing transcriptional activity corroborates the assumption of its role not only as a substrate but also as an inducer of the cytoprotective HO-1 system [25]. 

Free heme promotes Nrf2 activation by negatively modulating Bach1 function resulting in its translocation from nucleus to cytoplasm; simultaneously heme contributes to an increase in ROS levels which causes Keap1 detachment from Nrf2 permitting its translocation into nucleus where it binds ARE sites to regulate transcription of several antioxidant genes, including *HOMX1*. The figure also reports the heme metabolic fate, highlighting beneficial effects of the enzymatic reaction end-products.

## 3. Role of Heme End-Products

HO binds heme in a specific portion of its pocket to form enzyme-substrate complex, resulting in ferric heme-iron reduction to a ferrous state by the electron donated from the NADPH-cytochrome P450 reductase. Then, molecular oxygen binds with the complex to form a metastable oxy-form. The iron-bound oxygen converts to a hydroperoxide intermediate (Fe3^+^-OOH), and lastly the terminal oxygen of Fe3^+^-OOH attacks the α-meso-carbon of the porphyrin ring to form ferric α-meso-hydroxyheme. The latter reacts with molecular oxygen, leading to ferrous verdoheme-HO complex and CO formation. Conversion of ferrous verdoheme to ferric biliverdin–iron chelate requires oxygen and reducing equivalents. Ferric biliverdin iron is further reduced to the ferrous state by the reductase. The BV generated in the HO reaction is reduced to bilirubin-Ixα by the biliverdin reductase [26].

Bilirubin is a tetrapyrrole pigment mostly produced in the spleen following hemoglobin degradation by HO-1 first and biliverdin reductase then. It can be detected in different chemical forms in the blood, namely, conjugated with glucuronic acid (direct bilirubin), unconjugated bound to serum albumin (indirect bilirubin) and unconjugated unbound to serum albumin (free bilirubin) [27].

Both BV and BR as reducing species are powerful antioxidants and it has been shown by different studies cellular protective effects against injurious stimuli in vivo and in vitro. Bilirubin was generally regarded as a potentially cytotoxic waste product of heme demolition that needs to be excreted. However, it was shown that BR, at micromolar concentrations in vitro, acts as radical scavenger [28].

Nowadays BR is known to possess antioxidant and anti-inflammatory activity, and it is a biomarker for liver function; low levels of BR are correlated to cardiovascular risk, such as ischemic heart disease, hypertension, but also to diabetes type II, metabolic syndrome, and obesity [29,30].

Bilirubin major protective effect results from the biliverdin-bilirubin cycle mechanism that implies BR to react with oxygen radicals generating BV, which is then reduced back to BR by biliverdin reductase with the simultaneous consumption of NADPH. Chemical evidence of this mechanism was shown by McDonagh A. in his report where it is observed the formation of BV during incubation of bilirubin-albumin with 2,2’-azobis(2-amidinopropane) hydrochloride (AAPH) in vitro, suggesting that BV produced in the presence of albumin is not formed by the reaction of BR with alkyl peroxyl radicals, as previously assumed [31]. In particular, HO-1-derived BR is an efficient scavenger of reactive oxygen and nitrogen species (RONS). BR may also be relevant to prevent oxidant-mediated vasoconstrictive effects of TNF and angiotensin II, indeed inhibition of NADPH oxidase and protein kinase (PKC) activity by HO-1-derived BR reduce angiotensin II-mediated vascular injury [32].

Carbon monoxide (CO) has been identified as an important second messenger in the central nervous system (CNS) and it is known its ability to shutdown endothelial cell apoptosis activating the p38 MAPK pathway. Over the years CO multiple roles inside the complex human system have been discovered, one for all acting as a vasodilator via stimulation of soluble guanylate cyclase (sGC) [33]. HO is considered the main source of endogenous CO production.

Recently, CO significance in cellular regulated cell death programs (RCD) has been disclosed, pointing out its implication in autophagy, apoptosis, and pyroptosis modulation, ascribable to its ability to inhibit p38MAPK and NLRP3-ASC-mediated caspase 1 activation [34].

Exogenous CO can exert influence on inflammatory, proliferative, and apoptotic cellular programs. In particular, CO can modulate the production of pro- or anti-inflammatory cytokines and other mediators. HO-1 derived CO also seems to have immunomodulatory effects, probably due to its involvement in the antigen-presenting cells, T-cells, and dendritic cells functions regulation [9,35].

Iron is vital for cell growth and metabolism, but excess cellular iron is toxic because it donates electrons to produce hydroxyl radicals via the Fenton reaction. The ability of ferritin to sequester iron provides the dual function of segregating iron in a non-toxic form, as well as iron storage [36].

## 4. Protective and Detrimental Role of HO-1

The heme oxygenases (HO) other than provide for heme degradation also participate in cellular defense. It was observed that the activation of HO-1 is a ubiquitous cellular response to oxidative stress and the metabolites resulting from its participation in heme degradation are considered important antioxidant, as already mentioned before. Furthermore, iron released from HO activity stimulates ferritin synthesis, which ultimately contributes to iron detoxification that may be accountable for long-term cytoprotection noticed after HO induction. On the other hand, it has been underestimated the potential pro-oxidant effect of HO activity that may be reconducted to iron excessive release, involving deleterious reactions of iron reutilization and sequestration pathways. Indeed, the induction of HO activity may show both pro- and antioxidant consequences depending on redox homeostasis and iron metabolic fate [37].

HO-1 is generally considered a fundamental survival enzyme; only two cases of *HMOX1* gene deficiency were found in humans, showing phenotypes featuring chronic inflammation, asplenia, anemia, nephropathy, growth retardation, vascular injury, and tissue iron accumulation [38,39]. Notably, new studies have recently reported other cases of HO-1 deficiency in humans showing similar features and a rapid deterioration after symptoms onset, indicating HO fundamental role in organs protection from inflammation and oxidative stress, and thereby survival [40]. *HMOX1* serves as a protective gene under the anti-inflammatory, anti-apoptotic, and anti-proliferative actions of one or more of its end-products [2]. Moreover, HO-1 induction has shown many beneficial effects in a number of cells and tissues, and it is a well-known strategy cells put in place to overcome excessive stress consequences. The specific induction of this particular enzyme may be a potential advantageous approach against several disorders triggered by oxidative imbalance and inappropriate immune response. Interestingly, synthesis and discovery of new non-cytotoxic HO-1 inducers could be an innovative strategy to ameliorate oxidative damage and inflammatory conditions [41].

However, increasing evidence has shown a dark side of HO-1, acting as a critical mediator in ferroptosis induction and serving as induction factor for several diseases’ advancement. More and more studies are drawing the attention to HO-1 increased expression leading to onset and development of diseases, such as neurodegeneration and carcinogenesis [42].

Due to HO-1 implication in several malignancies, it may represent a new target for innovative therapeutic approaches. Bioactive compounds and new synthetized molecules with modulating activity towards HO-1 expression and consequently its activation could be a plausible path for cancer treatment [43].

It has been shown that several chemical drugs and natural bioactive extracts are able to induce HO-1 expression, such as curcumin, flavonoids and isothiocyanates, some of these are already being used and consist of food, medicinal plants, spices, and flavoring agents. The targeted induction of HO-1, in particular by non-toxic inducers, has been confirmed as an effective therapy for stress-related diseases [42,44,45,46].

Ultimately, granting protection to cells from oxidative stress HO-1 increases cell survival and suppress apoptotic programs activation. This kind of outcomes may lead to uncontrolled cells proliferation and induce carcinogenesis, metastasis and other ailments, such as neurodystrophic disorders. Because of these new lines of evidence HO-1 interest has been growing rapidly [47].

The cytoprotective effect of HO-1 was first revealed by the work of R. Tyrrell and colleagues, they proved that HO-1 induction is responsible for an adaptive cytoprotective effect in cultured human fibroblasts under oxidative stress conditions [48,49]. Thereafter, G. Balla and colleagues reported that HO-1 induction exerted the same function in cultured endothelial cells in response to oxidative injury [50]. All these studies though were not able to describe the actual HO-1 mechanism of action responsible for his cytoprotective effects, instead it became possible to address the molecular mechanism involved in programmed apoptosis mediated by HO-1 induction and its protective role in this context. Several studies unraveled a link between apoptotic programs triggering and HO-1 function, either in a direct way as the case reported by P. Soares et al., where they revealed HO-1 protective activity on endothelial cells exerted by avoiding cells undergoing TNF-mediated apoptosis, or by means of HO-1 end products, particularly CO, as it was observed that its exogenous administration to endothelial cells in vitro seems to stimulate the cytoprotective processes mediated by HO-1 activation. Other findings show that when CO is applied exogenously to endothelial cells in vitro, it can simulate the cytoprotective effect of HO-1 suggesting the conclusion that cytoprotective effect of HO-1 is mediated by carbon monoxide [51,52]. Kim et al., demonstrated that CO induces Nrf2-dependent HO-1 expression via Protein Kinase R-like ER Kinase (PERK) pathway and inhibits apoptosis activated by ER stress via p38 MAPK-dependent inhibition of C/EBP homologous protein (CHOP) expression [53].

Additionally, further observations of pharmacological inhibition of the p38α and p38β MAPK isoforms resulting in the suppression of HO-1 antiapoptotic effect proves the crucial involvement of HO-1in MAPK pathway [54,55].

## 5. Correlation between HO-1/Nrf2 Antioxidant Axis and GSH Biosynthesis

Although HO-1 antioxidant effects are primary reconducted to the redox cycle involving BV and BR, recent evidence shed light on the potential role played by another heme byproduct, carbon monoxide (CO). HO-1 provides for almost the totality of CO endogenous production via heme catabolism and for years researchers have been trying to uncover the mechanism underlying CO antioxidant effects reported in different studies [56,57]. CO derived from increased HO activity is responsible for the up-regulation of glutamate-cysteine ligase expression and restoration of GSH [58].

Specifically, the enzymes responsible for GSH synthesis, such as glutamate-cysteine ligase modifier subunit (GCLM) and the glutamate-cysteine catalytic subunit (GCLC), and enzymes involved into GSH utilization, such as glutathione reductase, glutathione peroxidase and glutathione S transferase (GST), are transcriptionally regulated, as well as HO-1, by Nrf2 [59,60].

Elevated GSH levels are observed in various types of cancer, and this makes the neoplastic tissues more resistant to chemotherapy. Depletion of GSH levels in A549 cells caused an increase in sensitivity to cisplatin, confirming its importance in determining drug-resistance [59]. Moreover, Furfaro et al., demonstrated that BSO, a selective and irreversible inhibitor of γ-glutamylcysteine synthetase (γ-GCS), and the inhibition of HO-1 are able to increase the sensitivity of neuroblastoma cells to the antitumor agent etoposide [61,62].

Increasing evidence point out CO role in the regulation of energetic metabolism. Even if it has already been established the inhibitory effect of carbon monoxide on the respiratory chain and its modulating effect on mitochondrial function [63,64,65,66] it is still unclear the mechanism behind it. Several reports have also assumed that CO protective effect against oxidative insults must be related to inhibition of cytochrome c oxidase, the terminal enzyme within complex IV of the Electron Transport Chain (ETC), by competitive binding oxygen resulting in a significant temporary burst of mitochondrial ROS production [67,68]. However, new findings propose an action mechanism more tending to other antioxidant scenarios.

Experiments with CO-releasing molecules (CORM) have been conducted to investigate the impact of CO, whether administrated via CORMs or endogenously produced by heme catabolism, on metabolic pathways. The use of CORMs facilitates the studies on CO intracellular interactions as it is an excellent source of controlled amount of CO. Averilla et al., experiments conducted on human keratinocytes lead the way to discover novel action mechanism of heme-derived CO as antioxidant response to different stimuli like oxidative UVB-induced damage to skin cells, suggesting an involvement of HO-1/CO signaling [69].

Furthermore, Kaczara and colleagues’ studies with CORM-401 showed accelerated pentose phosphate pathway (PPP), increased NADPH concentration and increased ratio of reduced to oxidized glutathione (GSH/GSSG), suggesting CO role as a fine regulator in the complex intracellular signaling system and underlining its cytoprotective effect [70].

The elevated production of reduction equivalents like NADPH, which is a cofactor required for GSH reduction, helps maintaining the reduced form serving as antioxidant and detox factor.

GSH is a cysteine-containing tripeptide and it is considered the main redox buffer in cells; it plays pivotal roles as cytoprotecting factor against a wide range of free radicals including reactive oxygen species (ROS), lipid hydroperoxides (LOOH), xenobiotic toxicants, and heavy metals. Among different redox couples found in the cell balancing the redox state, undoubtedly GSSG/2GSH couple is the most abundant [71].

Cells in a healthy state maintain a ratio of reduced to oxidized GSH approximately to 9:1 and a decreased value is considered as an oxidative stress index [72].

New lines of evidence seem to show a link between GSH depletion and ferroptosis. GSH functions as a direct antioxidant, as well as an important substrate for antioxidant GPX4 to prevent lipid ROS accumulation; thus, GSH depletion ultimately leads to increased lipid peroxidation. Under such circumstances, the homeostatic control of the steady state between LOOH formation and reduction is lost, lipid peroxidation is shown to be increased and consequently ferroptosis is initiated [73]. Thus GSH depletion has been classified as type 1 ferroptosis inducer [74].

## 6. HO-1 and Ferroptosis

HO-1 is critically involved in iron detoxification and ROS homeostasis and shows an ambivalent function in ferroptosis, beneficial and detrimental. Recently this new type of programmed cell death has drawn attention to redox systems protagonists as modulators of proliferation. Ferroptosis has been identified as a lipid peroxidation- and iron accumulation-dependent cell death associated to characteristic features as mitochondria dimensions reduction and enhanced ROS production which differentiate this process from other well-known regulated cell deaths as apoptosis or autophagy [75,76].

Due to its dual function in maintaining iron and ROS balance, HO-1 was suggested as a factor in ferroptosis by several studies [77,78,79,80,81]. Pharmacological inhibition and gene silencing of HO-1 also confirm its implication as trigger for ferroptosis increasing free-iron accumulation, and subsequently causing massive ROS production and lipid peroxidation. It appears that HO-1 selectively tend to one process rather than the other, specifically cytoprotection or induced cell, depending on redox misbalance between ROS generation and cell buffer redox capacity leading to an increase in the oxidative damage. One simple explanation could be that depending on the extent of stimuli to which cells are exposed HO-1 activation exerts either a cytoprotective effect or becomes cytotoxic [82].

As date, some molecules, including heme, erastin/sorafenib/RSL3, magnesium isoglycyrrhizinate, and withaferin A, have been found capable of triggering ferroptosis. It was proved that heme is an important factor for activation and death of human platelets, exerting its functions as trigger for ferroptosis, indeed it was assumed the key role played by heme-mediated lipid peroxidation and ferroptosis correlated to hemolytic disorders. Several lines of evidence show that the HO-1-mediated release of iron from hemin and the following generation of hydroxyl radicals are responsible for lipid peroxidation [83].

As GSH depletion leads to ROS accumulation, erastin, sorafenib and ras-selective lethal small molecule 3 (RLS3) represent three valid ferroptosis inducers as they, respectively, operate as xCT inhibitors and glutathione peroxidase 4 (GPx4) inhibitor affecting directly and indirectly on GSH levels.

Oxidative stress triggers ferroptosis and GSH plays a crucial role in ferroptosis, since it represents the substrate for GPX4 and is indispensable for preventing ferroptosis [84]. In vitro ferroptosis is typically induced after cysteine starvation, GSH depletion and inhibition of GPX4 [85,86]. The compound RLS3 developed by Stockwell et al. [87] can directly bind and inactivate GPX4, increasing the production of lipid hydroperoxides [88,89].

In particular, RLS3 inactivates GPX4-mediated alkylation of the selenocysteine present at the reactive site [90]. Proteins involved in iron metabolism, such as transferrin, transferrin receptor 1, and ferroportin, as well as HO-1, have been implicated in the ferroptotic process [80,91,92]. Transcriptome Investigation by Li et al., highlighted the role of HO-1 in curcumin-induced ferroptosis [93].

Nrf2 is fundamental as it induces ferritin expression capable of chelating ferrous iron, avoiding ROS overload and its potential detrimental outcomes [75,94,95].

Different studies have highlighted HO-1 expression role in augmentation or at least modulation of anti-cancer-agent induced ferroptosis via iron accumulation and ROS production [78,79]. However, the role of Nrf2/HO-1 signaling pathway and the mechanisms of action involved are still unclear [96] because, even though several studies have shown augmented HO-1 expression in cancer cells undergoing ferroptosis, the molecular mechanisms regulating the process remains controversial. Indeed, Adedoyin et al., showed HO-1-deficient renal cells are highly sensitive to erastin- and RSL3-induced ferroptosis [77]. Furthermore, melatonin significantly reduced ferroptosis and improved osteogenesis via the Nrf2/HO-1 pathway [97] and, additionally, Li and colleagues assessed that selective inhibition of Keap1-Nrf2/HO-1 pathway significantly abolished the anti-ferroptotic functions of panaxydol in lipopolysaccharide-treated cells [98].

## 7. HO-1 Physiological Functions in Different Tissues

HO-1 as ubiquitous molecule can be found in most of human tissues, hence its involvement and participation in many processes regarding each of them. The physiological role of heme oxygenase in iron homeostasis and the phenotypic consequences are evident in the HO-1 deficient mice. Heterozygous mating mice showed partial pre-natal lethality of the homozygous offspring and homozygous mating pairs did not yield viable litters [99,100]. As far as is known, there is a lack of studies evaluating the role of HO-1 in cancer using CRISPR/Cas9 technology. However, Mucha et al., reported that HO-1 protein level in CRISPR/Cas9-derived cell clones was diminished, both in basal and hemin-induced conditions [101].

As much as concerns adipose tissue, Vanella et al., studies report HO-1 activity upstream of canonical Wnt signaling cascade highlighting HO-1 modulation of adipocyte phenotype by its ability to regulate the transcriptional factors involved in adipogenesis. Upregulation of HO-1 gene expression and increased HO activity showed a decrease in adipocyte differentiation associated to an increased number of small healthy lipid droplets. It was also observed decreased adipocyte hypertrophy and TNFα levels and increased expression of the genes involved in the canonical Wnt signaling cascade, all evidence supporting the theory of HO-1 implication in this important pathway [102].

As reported in other studies and in the review written by Berg and Scherer, HO-1 induction in vivo and in vitro results in an increase in pre-adipocytes, a reduction in the number of hypertrophic adipocytes, and an increase in small adipocyte and adiponectin levels [103]. Additionally, induction of HO-1 in adipocytes is linked to a decrease in pro-inflammatory cytokines, such as TNFα and IL-6, and an anti-obesity effect resulting from evidence of reduction in weight gain. These findings highlight the pivotal role and symbiotic relationship of HO-1 and adiponectin in the modulation of the metabolic syndrome phenotype [104,105].

Studies on inducers of HO-1 expression, such as osteoprotegerin (OPG), show how they can affect osteoblast differentiation by increasing HO-1 levels, pAKT, and eNOS. It was demonstrated that osteoblast differentiation is upregulated by HO-1 expression, re-establishing osteoblast markers and resulting in induction of OPG and osteocalcin, on the other hand HO-1 expression seems to negatively regulate adipocyte stem cell differentiation [106].

In hematopoietic cells, heme oxygenase level may play a crucial role during stem cell proliferation and differentiation. It has been hypothesized that a reduction in erythroid heme oxygenase activity may be an important feature of the differentiation process. Heme oxygenase activity is highest in the spleen followed by the liver and bone marrow [3].

Heme oxygenase-1 (HO-1) plays an important role in bone marrow stem cell differentiation [107]. During fracture repair, activation of HIF-1 and its target genes, VEGF and HO-1, regulate osteoblasto-genesis and bone reabsorption suggesting a role of HO-1 in bone metabolism [108].

The mechanism underlying the HO-1-specific cell effect on osteoblasts and adipocytes is yet to be examined. The observation of HO-1 implications in angiogenesis and cell growth have brought scientists to the conclusion that HO system may play a regulating role in osteoblast cell proliferation, supporting this theory data show HO-1 derived CO functions as a modulator in vascular calcification.

HO-1 induction resulted in increased osteoblast differentiation alongside reduced ROS formation, thereby permitting the restoration of osteoblastic markers. Induction of HO-1 is essential for the resultant increase in pAKT, pAMPK, peNOS levels, and NO bioavailability [109,110,111]. In order to clarify potential links between genes of our interest PETAL tool analysis was performed to allow users to find hidden interactions among significant proteins belonging to the same pathway, and other proteins within possible linked pathways. From a biological point of view, this depth search helps to retrieve certain interesting genes potentially hidden and involved in important biological and cellular process. In Figure 2, it is possible to observe the depth first analysis performed by PETAL tool [112] starting from *HMOX1* gene, with *NOS3* (eNOS) as target gene and a depth search level equal to 10. The full path obtained by the analysis reads as follows: *HMOX1* -> *BLVRA* -> *UGT2B11*… -> *CYP3A4* -> *CYP2C8* -> *PLA2G4B* -> *PRKCA* -> *MAPK7* -> *NFE2L2* -> *NOS3*.

## 8. Contribution of HO-1 Induction in Pathological Conditions

### 8.1. Renal System

It has been demonstrated that HO-1 expression has relevance in the pathogenesis of hypertension, diabetic kidney disease, and renal disease. Besides, HO-2 deficiency contributes to a diabetes-mediated increase in renal dysfunction. The kidney usually maintains high metabolic activity, so it is easy to imagine the existence of a strong correlation with oxidant species activity and production. Several studies aforementioned sustain HO importance in maintaining homeostasis under oxidative stress conditions, and its role in the pathophysiology of many organs [113]. As far as it concerns the renal system, HO-1 induction is observed in multiple kidney substructures in response to oxidative damage, including glomeruli, proximal tubules and renal interstitium, as well as in renal mononuclear phagocytes [114]. In most models of acute kidney injury (AKI), HO-1 is primary induced in renal proximal tubules, also HO-1 expression displayed in glomeruli in models of diabetes and in infiltrating macrophages in acute renal transplant rejection [115]. Renal tubuli are constantly exposed to oxidative agents and stress conditions potentially resulting in damage of the entire renal system, in this context HO-1 has a key role protecting these important structures and its deficiency leads to higher susceptibility to tubular epithelial injury, as shown in various animal models [114,116,117].

Several studies on these models suggest that HO-1 synthesis is localized in the tubular epithelial cells. Thereafter, researchers interest shifted to hematuria or proteinuria role in the induction of HO-1 within renal tubuli. It was revealed that hematuria functions as major oxidative stress condition being able to induce HO-1 expression [118].

### 8.2. Central Nervous System

Oxidative stress plays a pivotal role in delicate central nervous system homeostasis, especially during cerebral ischemia, which may lead to neuronal injury and cell death. It was demonstrated the correlation between redox imbalance and ischemic damage [119,120] opening the way for more studies focused on elucidating HO-1 role in this context and pointing out the potential of new therapeutic strategies using, as a target, the endogenous antioxidant system for the amelioration of prognosis in patients affected by cerebral infarction.

Many natural compounds activating Nrf2-HO-1 pathway, such as sulforaphane, Ginkgo biloba extracts, curcumin, polyphenols, and triterpenoids exert neuroprotective effects against strokes in animal models [121].

Several lines of evidence disclose the involvement of HO-1 as cytoprotective defense in the pathogenesis of various neurodegenerative diseases, including Alzheimer’s disease (AD), amyotrophic lateral sclerosis (ALS), Huntington’s disease (HD), and Parkinson’s disease (PD) [122,123,124,125,126,127]. AD, PD, HD, but also ALS and Friedreich’s ataxia (FRDA), are referred to as “protein conformational diseases”, which are characterized by dysfunctional aggregation of proteins in non-native conformations.

As a result of protein misfolding, intracellular aggregates cause ER dysfunction which leads to oxidative stress [128].

It is known that neuron cells are particularly vulnerable to oxidative damage because of their particular characteristics, including high polyunsaturated fatty acid content in membranes, elevated oxygen consumption, and limited antioxidant defense capacity. Taken together, all these findings make clear how vital and fundamental is the antioxidant system, in which participates HO-1, for these cells [129]. Moreover, HO-1 end products showed an important antioxidant activity on neurodegenerative diseases too [130].

Results reported by Mhillaj et al., confirmed CO and Bilirubin ability to reduce ROS formation induced by soluble amyloid-β-peptide and fibrillar amyloid-β-peptide, and demonstrated the capacity of CO and bilirubin to reduce growth rate of Aβ oligomers in AD [131].

In physiological conditions, HO-1 expression is confined to outpoured neuroglia and neurons, and there are many factors of stress that increase its induction both in neuronal and non-neuronal cells. In patients affected either with AD and PD were detected higher levels of HO-1 expression in astrocytes of the hippocampus and cerebral cortex, together with enhanced serum and plasma levels of HO-1 [132,133]. Additionally, significantly higher HO-1 levels were observed in the plasma of patients in the early stage of PD and increased levels of HO-1 were associated with a reduced right hippocampal volume. Data regarding HO-1 induction in AD and PD may represent a warning sign of cognitive impairment [134,135]. In many neurodegenerative disorders hyperactivation of HO-1 leads to iron accumulation resulting in mitochondrial associated non-transferrin iron sequestration and macroautophagy, these conditions can be correlated to bioenergetic failure documented in Alzheimer disease and Parkinson disease [136].

Recent investigations draw the attention on HO-1 role within dopaminergic pathways, it was shown its implication in dopaminergic injury following polychlorinated biphenyls exposure suggesting a relevant contribution to PD pathophysiology. As a result to oxidative stress conditions dopaminergic cells exhibited a persistent expression of HO-1 [137]. Evidence from in vivo and in vitro studies reveals that enhanced HO-1 activity may either improve or exasperate neural damage, depending on the specific model employed, moreover it can affect duration and intensity of enzyme induction, and the chemical equilibrium of the redox microenvironment [10].

Aside from all the beneficial effects of enhancing HO-1 activity in central nervous system it is noteworthy highlighting the potential detrimental role of heme-derived carbon monoxide and iron that may contribute to increase intracellular oxidative stress and exacerbate disease progression [138].

### 8.3. Cardiovascular System

Oxidized low-density lipoproteins (LDL) play a crucial role in atherogenesis and are linked to antioxidant HO-1 system, actually LDL function as inducer of HO-1 expression. It was observed a correlation between serum bilirubin levels and decreased risks of CVD with protection against diabetes and vascular dysfunction [139]. HO-1 affects cardiovascular system via one of the heme degradation end-product, carbon monoxide (CO) which activates soluble guanylate cyclase and elevates cGMP causing vasodilation. Moreover, CO inhibits platelet aggregation, proliferation of vascular smooth muscle cells and apoptosis and concurrently stimulates angiogenesis. Several studies show how both over- and under-expression of HO-1 may contribute to pathogenesis of arterial hypertension. New promising strategies to ameliorate cardiovascular symptoms and pathological progression include pharmacological and genetic induction of HO-1, as well as the delivery of exogenous CO [140].

### 8.4. Inflammation

It is known that HO-1 and its byproducts have strong anti-inflammatory effects and new lines of evidence suggest the therapeutic potential of targeting HO-1 in several diseases set in the wide context of inflammation, such as inflammatory arthritis [141], psoriasis [142], neuroinflammation [143], and inflammatory bowel disease [144,145].

Experiments conducted on HO-1 deficient mice show an increased degree of inflammation and following transfection with HO-1 it was displayed an increase in anti-inflammatory effects, which support the theory of anti-inflammatory role of HO-1. The mechanism put in action by HO-1 is probably dependent on STAT3 since HO-1 is able to enhance STAT3 phosphorylation and further studies proved HO-1 anti-inflammatory effect loss in the presence of STAT3 siRNA or in STAT3-deficient mice [146,147,148]. However, STAT3 can exert its function upstream of HO-1, as it has been reported that STAT3 is required for the induction of HO-1 expression by interleukin-10 (IL-10), IL-6, and hyperoxia, suggesting the presence of a positive feedback loop between STAT3 and HO-1 [149]. Moreover, the protective role of HO-1 and CO on inflammation was proven in several disease models including ethanol-induced liver cell death, obesity induced liver fibrosis and lipid uptake increase, oxHDL and endothelial dysfunction in women with obesity [150,151,152].

## 9. HO-1 and Cancer

HO-1 primary role is surely its enzymatic activity on heme catabolism resulting in the production of heme metabolites, namely iron, CO, BV, and BR by which it exerts relevant anti-oxidative and anti-inflammatory functions. Many pathologies are characterized by an increase in HO-1 levels, including cancers.

HO-1 implication in tumorigenesis and cancer progression has been noted by many studies, thus the hypothetical role of the enzyme in cells proliferation and survival [153,154,155]. Moreover, HO-1 induces angiogenesis by modulating angiogenic factors’ expression resulting in a flourishing environment for cells proliferation and cancer progression. HO-1 translocation in nucleus was attested in cancerous cells where it allows tumor mass growth and spread, showing a function disentangled from its enzymatic activity. HO-1 nuclear presence has also been noticed in chronic myeloid leukemia cells resistant to imatinib treatment [156,157].

Several reports showed that HO-1 basal promoter activity and the induction level of HO-1 gene, in response to stress, are dependent on the (GT)n repeat number. It has been demonstrated that shorter (GT)n repeats within the *HMOX1* promoter results in higher transcriptional activity and it is associated with increased risk of pancreatic [158] and gastric cancer [159]. Conversely, the long (GT)n repeats in *HMOX1* gene promoter were reported to correlate with a higher risk of breast cancer, esophageal squamous cell carcinoma, and laryngeal squamous cell carcinoma [160,161,162].

HO-1 overexpression is generally observed in different human neoplastic diseases, among others colon [163], renal [164], gastric [165], prostate [166], bladder [167], glioma [168], oral [169], breast [170], and lung [171] cancers. HO-1 induction by various stimuli, such as cytokines and growth factors release together with enhanced stress conditions, was shown to be related with activation of several signaling cascade and engagement of transcriptional factors as NF Kb, AP2, and Nrf2.

Cancer cells express elevated ROS levels as a result of abnormally rapid proliferation and fast metabolic rate which enhances tumorigenesis. It appears that high ROS levels induce cellular stress which may affect cancer cells stability and make them more vulnerable to oxidative stress-induced cell death. Nevertheless, cancer cells have also shown a tendency in developing redox adaptation by promotion of antioxidant and anti-apoptotic systems, enhancing cell survival and chemoresistance [172,173].

Recent studies have reported *NRF2* and *KEAP1* mutations in human cancers. It seems clear that HO-1 has a key role in antioxidant response as it is known to be one of the main targets of Nrf2 activation, and also may be implicated in carcinogenesis as Nrf2 promotes proliferation processes. [174,175,176]

A sustained Nrf2 pathway activation may lead to the creation of a favorable environment for tumor cells growth and survival by maintaining an appropriate ROS levels balance. Nrf2 accumulation and protracted activation seems to create advantageous conditions for premalignant or cancerous cells not only to proliferate but to escape cell death controls check points and apoptosis leading to a facilitated way for metastasis spread and chemoresistance. Moreover, oncological patients presenting elevated level of Nrf2 expression in their tumor generally show a lower survival rate. Therefore, Nrf2 is considered a prognostic molecular marker for determining the status of cancer progression [177,178].

It is known that HO-1 does not possesses a DNA binding domain, so the mechanism behind HO-1 involvement in genes transcriptional processes and its correlation to cancer progression is still unclear and requires further investigation. Modulation of immune regulatory functions of myeloid cells has been linked to HO-1 expression, which is implicated in reducing proinflammatory cytokines release, including tumor necrosis factor-α, and concurrently enhancing immunosuppressive cytokine upregulation, such as interleukin-10 (IL-10). More evidence shows HO-1 promotion of inflammation-associated angiogenesis due to its ability up-regulating VEGF expression in macrophages (Figure 3) [179]. Studies on endothelial cells proliferation showed a link between HO-1 overexpression, NO, and VEGF [180,181].

HO-1 has an impact not only in the innate immunity, but it also contributes to adaptive immune response during tumor development.

Andersen and colleagues conducted a study whose results showed that immunosuppressive HO-1-specific CD8^+^ regulatory T cells were found in patients tumor tissues and peripheral blood, hence the HO-1 intrinsic immunosuppressive activity which could be studied as target for therapeutic intervention through immunomodulation [182].

HO-1 exert its cytoprotective role in early stages of tumorigenesis, but it then stimulates malignant cells growth, proliferation, and invasion of long term. Alongside its role in cancer cell proliferation, HO-1 is capable of promoting the progression of neoplasms by influencing or affecting tumor microenvironment [156].

A recent study by Yamamoto et al., showed HO-1 vicarious contribution, through its end-product CO, to cancer cells adaptation to oxidative stress condition and drugs through the inhibition of heme-containing cystathionine β-synthase, resulting in decreased methylation of PFKFB3, an enzyme producing fructose 2,6-bisphosphate (F-2,6-BP), and shift of glucose metabolism to the pentose phosphate pathway (PPP), causing a subsequent increase in NADPH to restore reduced GSH pool [183].

This shift to the PPP metabolism has been commonly seen in cancer cells but it was primary linked to the activation of different antioxidant response systems disentangled from HO-1, TIGAR activation is an example. In particular, TP53-induced glycolysis and apoptosis regulator (TIGAR) is a p53 target that functions as a fructose-2,6-bisphosphatase, thereby it is able to regulate energy metabolism, oxidative stress, and amino acid metabolism through balancing glycolysis and oxidative phosphorylation, as well as autophagy [184,185]. TIGAR ability to decrease intracellular ROS levels also plays a role in the p53 cytoprotective activity. p53 can promote through different genes an increased generation of ROS, which can lead to apoptosis or else get in a feedback loop in which ROS can be responsible of further p53 activation. TIGAR may play an active role in DNA repair mechanism, other than decreasing ROS levels and allowing cell survival, due to the generation of intermediate metabolites of the pentose phosphate pathway (PPP), including NADPH and ribose-5-phosphate, which are important precursors of DNA biosynthesis and repair [186,187,188].

However, different studies suggest that TIGAR expression and activity can be detached from the p53 response and its contribution to cancer development may depend on the regulation mechanism behind it. Even if TIGAR’s role in promoting cancer development might appear in conflict with what we would expect it to function in the p53 tumor suppressor pathway, it is important to notice that in tumor cells overexpressing TIGAR, its expression is uncoupled from p53 expression. Indeed, further analysis in tumor cell lines prove that the basal expression of TIGAR is not directly dependent on the maintenance of wild type p53 [189]. Nowadays several consequences of TIGAR activity can be predicted, including the preferential shift to alternative metabolic fates, such as the oxidative or non-oxidative branches of the pentose phosphate pathway (PPP). The PPP plays a key role in generating NADPH, which functions as antioxidant and is implicated in fatty acid synthesis [190,191].

Having regard to the findings abovementioned, and the recent notion of heme-derived CO implications in determining a metabolic shift towards the pentose phosphate pathway (PPP), we could venture the hypothesis of a correlation between these two important antioxidant systems exerting their activity simultaneously in order for the cell to replenish its reductants and GSH sources, and provide for an appropriate response to oxidative stress condition, both in physiological and pathological state. Figure 4 shows a depth analysis performed by PETAL tool [112] starting from *HMOX1* gene, with *TIGAR* as target gene and a depth search level equal to 10. The full path obtained by the analysis reads as follows: *HMOX1* -> *BLVRA* -> *UGT2B11*… -> *CYP3A4* -> *PLA2G4B* -> *PRKCA* -> *PTK2* -> *MAPK1* -> *TP53* -> *TIGAR*.

## 10. Therapeutical Strategies and New Perspectives—HO-1 Modulation

In the past few years, it has been shown an increasing interest towards HO system as a potential pharmacological target, therefore researchers have focused their work on developing drugs able to modulate both Heme Oxygenase’s expression and activity, drawing the attention mostly on the inducible isoform, HO-1.

According to the widely established cytoprotective role played by HO-1 in response to different pathological conditions, its upregulation could represent a valuable therapeutic strategy for the management of diseases as atherosclerosis, inflammation, vascular injury, hypoxia, and ischemia, as well as several metabolic dysfunctions [45,192]. HO-1 induction also seems to serve as a valid option for the treatment of ocular malignancies as glaucoma, diabetic retinopathy, and retinal degeneration [193,194].

Several studies have also assessed a significant reduction in CO levels, due to a decreased HO-1 activity, in women affected by preeclampsia. *HMOX1* activation by statins and the following increase in HO-1 expression could diminish preeclampsia symptoms, allowing to extend the pregnancies [195].

Despite the promising perspectives, therapeutic approach based on HO-1 induction remains a challenging path to pursue. Excessive activation of HO system may contribute to tissue damage development or it could be exploited by cancer cells as a defense mechanism enhancing tumor proliferative processes [196]. Hence, the propension of researchers to focus their attention mostly on HO-1 negative modulation.

HO-1 inhibition represents a valid pharmacological strategy for the treatment of conditions associated to significant enzyme expression and elevated levels of its byproducts. HO-1 inhibitors main field of application is embodied by cancer therapy, the reason behind this observation is evident looking at the several studies pointing out HO-1 cytoprotective and anti-apoptotic effects, which induce carcinogenesis and chemoresistance [154].

The first therapeutic approach to regulate HO activity began with the development of molecules capable of downregulating HO activity by competitive inhibition. When finally results were obtained the primary role played by these compounds was the clinical use in neonatology for newborns with hyperbilirubinemia [139].

The first molecules used as HO-1 inhibitors were the metal-porphyrins (MP) which are protoporphyrin IX analogs presenting another metal instead of the central iron atom, from this substitution were obtained zinc protoporphyrins (ZnPPIX), tin protoporphyrin (SnPPIX), and chromium protoporphyrin (CrPPIX). Their chemical structure allows MPs to interact with HO-1 heme binding pocket and affect enzyme activity [197].

Studies with synthetic metal-porphyrin complexes having the central iron atom replaced by other metals showed the inability of metabolic enzymes to eliminate these derivates, suggesting novel and still unknown biological properties potentially useful both in clinical and experimental fields. Such studies have revealed the impact of the central metal atom in the physiological and pharmacological activity of metal-porphyrin complexes [198]. Several experimental results confirmed MPs ability to reduce HO-1 activity, in particular SnPPIX treatment was observed to attenuate Kaposi sarcoma cells growth [199]. Further data showed an improvement in cisplatin efficacy following ZnPPIX inhibition of HO-1 in liver cancer, both in vivo and in vitro [200]. Even though MPs represent the first generation of HO-1 inhibitors their strong chemical similarity with heme permits to interfere with other heme-containing enzymes, thus the multiple side effects result from the lack of selective action towards HO-1 isoform, but also from the effect exerted by MPs on others heme-dependent enzymes, such as Nitric oxide synthases (NOS), soluble guanylyl cyclase (sGC) and cytochrome P450 [201].

The increasing data collection on HO-1 and the occurrence of the beforementioned side effects have redirected scientific research to develop new inhibitors presenting a non-porphyrin structure. Imidazole-based inhibitor showed anti-proliferative effects in several cancer cell lines [202,203]. This class of compounds is commonly characterized by a peculiar non-competitive binding mode in which heme ferrous iron oxidation and O_2_ binding are prevented when heme is placed in the HO-1 binding pocket [204,205].

The prototype is represented by azalanstat, an imidazole-dioxalane compound [206]. Based on the azalanstat structure it was possible to synthetize several compounds showing a good degree of selectivity for the HO-1 isoform [207]. Imadazole-dioxalane derivatives are classified as non-competitive inhibitors because they bind an HO-1 site which is not the catalytic one [208]. In vitro studies proved the antitumoral efficacy of these compounds in breast cancer cells [209] and in prostate cancer cells [210].

Recently, HO-1 inhibition has been evaluated via gene silencing, using RNA interference (RNAi) technique. The use of small interfering RNA (siRNA) or short harpin RNA (shRNA) specific for HO-1-coding mRNA induced apoptosis in lung cancer [211], colon carcinoma [212], and leukemic cells [213].

Even though it was proved the impact of HO-1 modulation in cancer cell proliferation, obviously the enzymatic inhibition cannot be considered as the only practicable treatment strategy for cancer therapy, but it represents a therapeutic option, as adjuvant, in order to increase tumor cells sensibility to chemo and radiotherapy [202].

In conclusion, the Heme Oxygenase-1 plays a significant role in multiple biological systems, both known and yet to be discovered; this encourages researchers’ investigations and further analyses to move from basic science to clinical application.

## Figures and Tables

**Figure 1 biomolecules-11-00589-f001:**
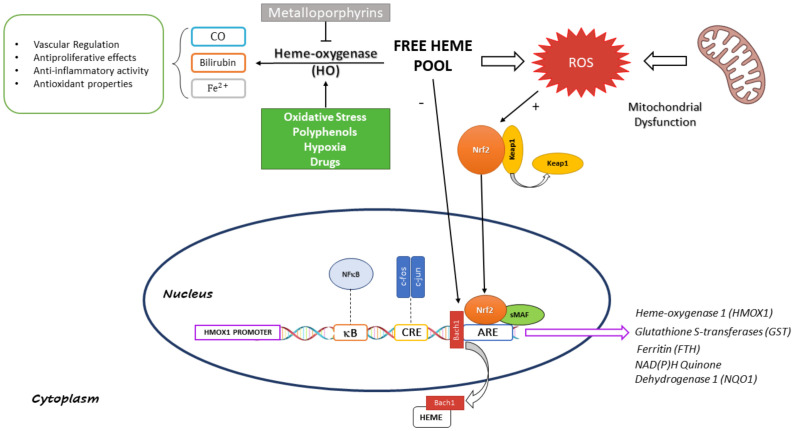
Schematic Representation of Heme-Oxygenase 1 Transcriptional Regulation and Function.

**Figure 2 biomolecules-11-00589-f002:**
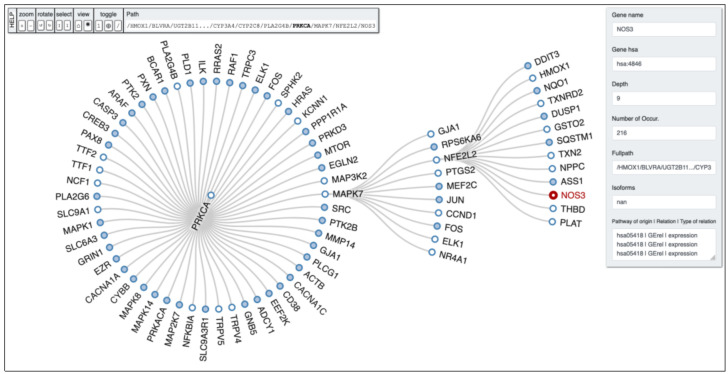
PETAL output tree obtained by running the depth-first search algorithm with a depth equal to 10 retrieved from the KEGG database (hsa00860) with *NOS3* as target gene.

**Figure 3 biomolecules-11-00589-f003:**
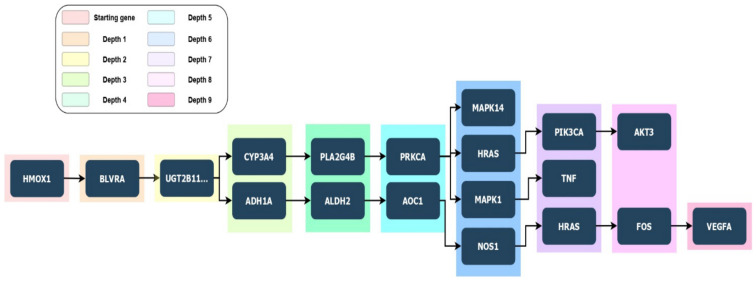
A schematic form of a depth tree coming from PETAL data results. It consists of a root node (*HMOX1* as starting gene) and subtrees connected to the root. Each subtree consists of several nodes (genes) leading to the target gene. The target genes detected by PETAL tool analysis are the following ones: (i) *MAPK14* (depth 6), (ii) *AKT3* (depth 8), (iii) *TNF* (depth 7), and iv) *VEGFA* (depth 9).

**Figure 4 biomolecules-11-00589-f004:**
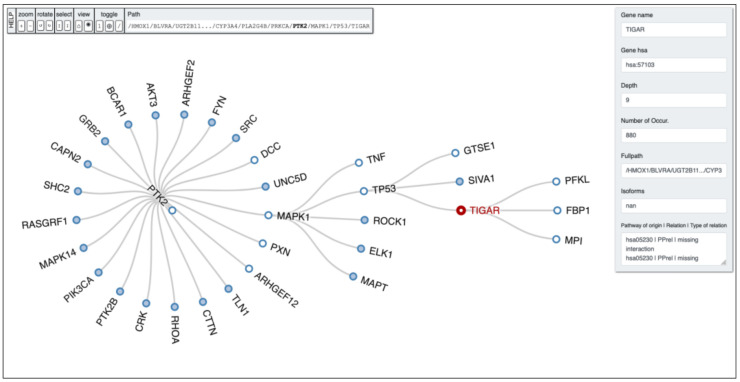
PETAL output tree obtained by running the depth-first search algorithm with a depth equal to 10 retrieved from the KEGG database (hsa00860) with *TIGAR* as target gene.

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
