# Peer review of "Heme Oxygenase-1 Signaling and Redox Homeostasis in Physiopathological Conditions"

_biomolecules, 2021, doi:10.3390/biom11040589_

Round 1

Reviewer 1 Report

General comment:

This manuscript, entitled “Heme oxygenase signaling and redox homeostasis in physio-pathological conditions,” authored by Consoli et al. reviewed the heme oxygenase-I role in the biological pathway and potential therapeutical strategies to investigate this antioxidant system. This review has updated information to cover most of the introduction of heme oxygenase 1, regulation, physiological function etc. In my opinion, this is a valuable work and is suitable for publication in biomolecules after the authors have addressed the following comments and questions:

Specific comments:

  • What is the fate of free iron and the role of iron storage protein? A short description may be needed.
  • What is the role of HO-1 gene associated between polymorphism and cancer proliferation and progression?
  • What inhibits HO-1 functioning? Are there endogenous or exogenous inhibitors other than metalloporphyrins?
  • How the depth analysis of HO-1 gene to the target gene is helpful in this kind of study? Why PETAL tool is the preference here? The author used this tool several times, and a description, in short, will help to understand the results for new readers.
  • HO-1 is ubiquitous; knocking out this gene is deleterious? If not, what are the alternative path?

Author Response

We thank all the reviewers for these valuable comments and suggestions. Comments are all valuable and very helpful to improve our manuscript. We clearly replied point-by-point to all reviewer concerns. Review has been extensively modified and improved.

All changes in the text are highlighted in yellow. Please find the author's responses below:

Specific comments:

What is the fate of free iron and the role of iron storage protein? A short description may be needed.

Iron is vital for cell growth and metabolism, but excess cellular iron is toxic because it donates electrons to produce hydroxyl radicals via the Fenton reaction. The ability of ferritin to sequester iron provides the dual function of segregating iron in a non-toxic form as well as iron storage. (11986201).

What is the role of HO-1 gene associated between polymorphism and cancer proliferation and progression?

We agree with the reviewer’s comment. Several reports showed that HO-1 basal promoter activity and the induction level of HO-1 gene, in response to stress, are dependent on the (GT)n repeat number. It has been demonstrated that shorter (GT)n repeats within the HMOX1 promoter results in higher transcriptional activity and it is associated with increased risk of pancreatic (21495030) and gastric cancer (18502573). Conversely, the long (GT)n repeats in HMOX1 gene promoter was reported to correlate with a higher risk of breast cancer, esophageal squamous cell carcinoma and laryngeal squamous cell carcinoma (19895178; 27400252; 17726138).

What inhibits HO-1 functioning? Are there endogenous or exogenous inhibitors other than metalloporphyrins?

Metalloporphyrins represent the first generation of HO-1 inhibitors. However, due to their strong chemical similarity with heme, they are able to interfere with other heme-containing enzymes. Imidazole-based inhibitor showed anti-proliferative effects in several cancer cell lines (29628790; 31276659). This class of compounds is commonly characterized by a peculiar non-competitive binding mode in which heme ferrous iron oxidation and O2 binding are prevented when heme is placed in the HO-1 binding pocket (23097500; 29783634).

How the depth analysis of HO-1 gene to the target gene is helpful in this kind of study? Why PETAL tool is the preference here? The author used this tool several times, and a description, in short, will help to understand the results for new readers.

PETAL tool allows users to find hidden interactions among significant proteins belonging to the same pathway and other proteins within possible linked pathways. From a biological point of view, this depth search helps to retrieve certain interesting genes potentially hidden and involved in important biological and cellular process.

HO-1 is ubiquitous; knocking out this gene is deleterious? If not, what are the alternative path?

The physiological role of heme oxygenase in iron homeostasis and the phenotypic consequences are evident in the HO-1 deficient mice. Heterozygous mating mice showed partial pre-natal lethality of the homozygous offspring and homozygous mating pairs did not yield viable litters (9380735; 9380736). As far as known, there is a lack of studies evaluating the role of HO-1 in cancer using CRISPR/Cas9 technology. However, Mucha et al., reported that HO-1 protein level in CRISPR/Cas9-derived cell clones was diminished, both in basal and hemin-induced conditions (29694447).

Reviewer 2 Report

The authors review Heme oxygenase signaling in physio-pathological conditions. Although the subject is widely explored, in several papers, the authors have done an ok job of bringing together the literature but have poorly written the article. Additionally, there is EXTENSIVE plagiarism detected in this paper. Understandably this is a review article but still the extent of copying entire paragraphs cannot be possible. The authors need to write them in their own words not copy paste from literature.

I would highly recommend that the authors use a plagiarism tool to check their paper.

Author Response

We thank all the reviewers for these valuable comments and suggestions. Comments are all valuable and very helpful to improve our manuscript. We clearly replied point-by-point to all reviewer concerns. Review has been extensively modified and improved.

All changes in the text are highlighted in yellow.

The authors review Heme oxygenase signaling in physio-pathological conditions. Although the subject is widely explored, in several papers, the authors have done an ok job of bringing together the literature but have poorly written the article. Additionally, there is EXTENSIVE plagiarism detected in this paper. Understandably this is a review article but still the extent of copying entire paragraphs cannot be possible. The authors need to write them in their own words not copy paste from literature.

I would highly recommend that the authors use a plagiarism tool to check their paper.

The manuscript was revised and improved as suggested

Reviewer 3 Report

In this review, Consoli and colleagues analyze the role of Heme Oxygenase-1 in normal and in pathologic conditions, with particular regards for cancer and neurodegenerative diseases. A perspective of the beneficial and detrimental roles played by the protein is given, including its relationship with relatively recently discovered cellular pathways, such as Ferroptosis. Beside these points of strength, the work of the authors is very ambitious, the topics covered by the review are very broad and the work does not keep its promises. Thus, a thorough and careful revision is needed, especially to correct some gross errors. In general, the text body is fragmented and reader risks to get lost among the sections, as a clear guiding thread is missing. On the one hand, some aspects are just introduced and then put apart. On the other hand, one passes from a topic to another without a clear rationale, often returning, in part or completely, to topics already addressed. Because of this, redundant or superfluous paragraphs are often present. Very frequently, proper reference for the contents reported by the authors is missing, and the need to standardize names and abbreviations is evident. Examples are given below, in conjunction with other specific requests, but the authors should take in account that only most evident missing references, redundant paragraphs and symbol requirements are outlined.

Specific concerns

1: Actually, very little is said about Heme Oxygenase 2, almost nothing on Heme Oxygenase 3 while the title of the paper generically refers to heme oxygenase. Maybe the title could be changed properly.

2: line54-57: is not clear why the authors insert here a paragraph briefly elucidating NRF2 ability to localize in the nucleus and then leave it apart without better clarifying this topic more in depth.

3: Line58: the paragraph should be extended and clarified properly or deleted.

4: lines79-82, it is not clear what the authors want to tell with this paragraph and where it should lead.

5: line86: there are other sMAFs able to interact with NRF2 and there is no reference to account for such a statement.

6: NRF2 is not the only transcription factor able to regulate HO-1 transcription. The authors in fig1 highlight this. Nevertheless, they are not mentioned in the text, neither is the regulation of their activity, on the contrary of what occurs for NRF2

7: In general, there is a huge need to uniform names and symbols (e.g. Nrf2, NRF2, nrfs2 etc.). The authors could take in account that gene or RNA transcripts symbols should be italicized and in capital letters, while proteins symbols should be in capital letters, in humans, not italicized.

8: Authors should not assume that every reader knows the molecular processes mediated by HO-1. Thus, could be of help if a brief recall of how heme end products are produced, is given.

9: it is not clear how HO-1 antioxidant activity is linked to GSH level implementation as stated in the title of the section. Also, it is not clear what the authors mean for “GSH levels implementation”.

10: Figure1 legend should be implemented, to explain what is reported in the figure properly.

11: Apoptotic pathway is very complex and explaining it in a paragraph (244-248) it is unlikely. Thus, the authors should decide whether a more in depth, general explanation of the apoptotic process is needed, or if it is sufficient just to focus on the part of the process that is influenced by HO-1 activity. Of note, in the way the authors wrote in the text, it is not clear how and at what level the activity of HO-1 influences the apoptotic process. In line with this, it seems that the whole paragraph (lines 244-251) is somehow superfluous, and for this reason the authors should better specify both the introductory part and, more importantly, the one concerning the action of HO-1

12: Caution has to be taken affirming that “several molecules have been found capable of triggering ferroptosis through HO-1 modulation” and references on how ferroptosis inducers modulate HO-1 expression should be given, as in many cases, this is not their main mechanism of action. In line with this, RLS3 is mainly an inhibitor of GPX4, which in turn uses GSH as cofactor. Thus, the way by which RLS3 can deplete GSH (line 279) should be better elucidated and referenced.

13: Recent reports highlight the activity of NRF2 and its role opposing the ferroptotic process. NRF2 transcriptional regulation is not only related to its ability of inducing ferritin transcription. Indeed, NRF2 is able to induce the transcription of most of the genes responsible of regulating the cellular defense against ROS and lipid peroxidation (which is one of the main triggers of the ferroptotic process), including the proteins responsible for GSH synthesis and GPX4 itself. Thus, NRF2 activity should be definitely better elucidated and if controversies are present (e.g. if NRF2 activity, through an excessive HO-1 synthesis, could induce ferroptosis) this should be discussed.

14: Authors state that HO-1 has a role in many neurodegenerative diseases, such as AD, ASL, HD and PD. Nevertheless, proper references are missing and a better evaluation of the positive or detrimental role played by HO-1 in these pathologies is required. Moreover, although could be difficult to cover all the neurodegeneration field, especially in regards of rare neurodegenerative disorders, HO-1 and NRF2-axis regulation is of fundamental importance in some of them and the discussion could take advantage of this, adding interest to an otherwise well-known topic in the literature which is abundantly covered by specific reviews.

15: Some parts in figures 2, 3 and 4 are difficult to read and could be easily enlarged to help the reader (e.g., the legend in figure 3)

16: NRF2 regulation of HO-1 expression is introduced in the Transcriptional regulation paragraph (starting from line 71), but its role is explained (and the name of the protein is re-introduced in full, without a reason) in line 385, in a subsection of the “Contribution of HO-1 Induction in Pathological Conditions”. Maybe this apparently evident fragmentation could be avoided or better circumstantiated.

Minor points

Line 28: a reference is missing.

Line 30: a reference is missing.

Line32: It would be better to use the symbol HO instead of heme oxygenase, as the symbol was introduced to this purpose.

Line35: Hsp32 symbol was used without introducing the protein full name. Notably, the protein full name was used below (line 37), in a sentence that is redundant with what is reported in line 35.

Line64-69 a couple of references are missing.

Line70: the paragraph content would be more coherent with a title as:  “HO-1 transcriptional regulation”.

Line71: NRF2 was already introduced and HMOX1 should be in italics if referred to HO-1 gene. Note that all gene symbols should be italicized.

Line90: Is Nrfs2 a typo?

Line107: there is a missing reference.

Line173: it is unclear why the authors introduce NADPH full name only here, as it is just reported above. In line with this, Glutathione, which is introduced for the first time at line 81, without abbreviation, is used multiple times with its full name, sometimes accompanied by its symbol (e.g. line 170 or 177) and almost never the symbol is used alone (line 144).

Line 183: reference is missing.

Lines 203-206 missing references.

Author Response

We thank all the reviewers for these valuable comments and suggestions. Comments are all valuable and very helpful to improve our manuscript. We clearly replied point-by-point to all reviewer concerns. Review has been extensively modified and improved.

All changes in the text are highlighted in yellow.

Please find attached the author's responses.

Round 2

Reviewer 2 Report

The authors have significantly improved the manuscript and addressed my concerns.

Reviewer 3 Report

Author addressed most of the comments made.